# The Compressive Strength and Microstructure of Alkali-Activated Mortars Utilizing By-Product-Based Binary-Blended Precursors

Otman M. M. Elbasir [1], Megat Azmi Megat Johari [2], Zainal Arifin Ahmad [3], Nuha S. Mashaan [4,*] and Abdalrhman Milad [5,*]

1   Department of Civil Engineering, High Institute of Science and Technology, Qasr Bin Ghashir 22131, Libya; othmanelbasir@hinstitute-bcv.edu.ly

2   Department School of Civil Engineering, Universiti Sains Malaysia, Gelugor 11700, Malaysia; cemamj@usm.my

3   School of Materials and Mineral Resources Engineering, Engineering Campus, Universiti Sains Malaysia, Nibong Tebal 14300, Malaysia; srzainal@usm.my

4   Department of Civil Engineering, School of Engineering, Edith Cowan University, 270 Joondalup Drive, Joondalup, WA 6027, Australia

5   Department of Civil and Environmental Engineering, College of Engineering, University of Nizwa, P.O. Box 33, Nizwa PC 616, Oman

*   Correspondence: n.mashaan@ecu.edu.au (N.S.M.); a.milad@unizwa.edu.om (A.M.)

**Abstract:** Researchers have investigated the feasibility of using ultrafine palm oil fuel ash (u-POFA) as a cement replacement material because of its potential to reduce the environmental impact of concrete production. u-POFA, a by-product of palm oil fuel combustion, is a suitable replacement for Portland cement in concrete mixes because of its sustainability and cost-effectiveness. This study investigated the microstructural and compressive strengths of alkali-activated mortars (AAMs) based on fly ash (FA) and granulated blast-furnace slag (GBFS) being added with varying percentages of u-POFA. The mixture samples were prepared in eighteen mortars using sodium metasilicate ($Na_2SiO_3$) as the source material and sodium hydroxide (NaOH) as the alkaline activator. This study used field-emission scanning electron microscopy coupled with energy-dispersive X-ray spectrometry, X-ray diffraction, X-ray fluorescence, and Fourier-transform infrared spectroscopy to characterize the binary-blended mortars after 28 days of curing and determined the strength of the FA+GBFS (87.80 MPa), u-POFA+GBFS (88.87 MPa), and u-POFA+FA mortars (54.82 MPa). The mortars' compressive strength was influenced by the $CaO/SiO_2$ and $SiO_2/Al_2O_3$ ratios in the mixture, which was directly due to the formation rate of geopolymer products of the calcium–alumina–silicate–hydrate (C–(A)–S–H), aluminosilicate (N–A–S–H), and calcium–silicate–hydrate (C–S–H) phases. Based on the contents of FA and GBFS, u-POFA significantly enhanced concrete strength; therefore, u-POFA used in a suitable proportion could enhance binary-blended AAMs' microstructure.

**Keywords:** ultrafine POFA; compressive strength; ground blast-furnace slag; geopolymer concrete

## 1. Introduction

### 1.1. Research Background

    Palm oil fuel ash (POFA) is a waste product from burning palm oil for energy. It is a fine, powdery material comprising a mixture of silica, calcium oxide, and other minerals. It is used as a soil amendment and fertilizer in agriculture, a construction material, and a fuel source in industry. In addition to reducing the amount of required cement, it has the beneficial properties of improving concrete strength, durability, and workability [1]. Zero-cement concrete is an improved version of geopolymer concrete, where the thermal activation of the binding materials, such as fly ash (FA) and granulated blast-furnace slag (GBFS), is combined with alkaline solutions, namely sodium hydroxide (NaOH) and sodium silicate

($Na_2SiO_3$). A full or partial replacement of steel slag can ensure durability and has economic benefits. Hence, the back polymerization reaction is pivotal in OPC-less concrete production [2]. The environmental consequences of carbon dioxide emission encourage the use of base materials, such as (FA), (u-POFA), and (GBFS), in developing alkali-activated and geopolymer concrete [3]. The stabilization and solidification of industrial solid waste with chemical activators produce new geo-polymeric binders, such as binary-blended, alkali-activated mortar (AAM). However, alkali-activated mortar is a useful product in industrial waste treatment due to the addition of CaO/SiO2 and SiO2/Al2O3 to the waste materials [4]. GBFS is a natural solid waste material used with other appropriate materials to produce blended AAM. GBFS alters the amount of amorphous Si, Al, and Ca, thus altering the mechanical properties of the resulting blended AAM. For example, mixing GBFS with fly-ash-based AAM at a constant FA/GBFS weight ratio affects the performance of the binary-blended AAM. The increase in FA of AAM strength is due to the incorporation of external amorphous Si via the addition of high-quality GBFS. The structure of GBSF determines its action and can be enhanced by varying the $SiO_2/AlO_3$ and Si/Ca ratios added to an unstructured source of Si [5].

### 1.2. Literature Review

Ultrafine palm oil fuel ash (u-POFA) is a by-product of palm oil fuel combustion. It is a fine, light-colored powder composed of silica, alumina, and other minerals. Researchers have conducted extensive investigations of the pozzolanic properties of u-POFA. Studies have shown that pulverized fuel ash (PFA) improves the strength and durability of concrete and mortar and reduces concrete permeability [6]. PFA also reduces the water absorption of concrete and mortar and concrete shrinkage. In addition to its pozzolanic properties, researchers have investigated the potential of using PFA as a cement replacement. Using PFA as a partial cement replacement in concrete and mortar improves its strength and durability. PFA also reduces the hydration heat of concrete and mortar and alkali–silica reaction [7].

In a previous study, a geopolymer mixture treated for 28 days had a strength of 51 MPa. The mixture was treated with a GBFS/FA hybridization ratio of 20/80 using $Na_2SiO_3$/NaOH of 1.5 and 40% alkali activator liquid [8]. Furthermore, u-POFA can be utilized as a raw material for blended geopolymers or alkali-activated synthesis. Southeast Asian countries, including Indonesia, Malaysia, and Thailand, have a rich source of POFA and can use it as a raw material for geopolymers or alkali-activated mortars [9]. The pre-treatment steps, such as sieving, calcination, and grinding, influence the effects of the waste materials utilized as raw materials for geopolymers [10]. Research is currently focused on the synthesis of blended geopolymers or alkaline-activated geopolymers from POFA and GBFS [6,11], but POFA and FA require improvement in their effectiveness. Si and Al provide POFA with good pozzolanic properties, which allow it to be used at 30% via binder weight replacement in PC concrete and up to 60% when being used to treat POFA [11]. POFA concrete has a high compressive strength of 104 MPa after 28 days of treatment. Recently, a group of researchers investigated using POFA as a complementary material blend with other aluminosilicate materials to produce geopolymer cement pastes or mortars [12]. For instance, blending low-calcium FA with POFA resulted in concrete with a compressive strength of 30 MPa after 28 days of curing [13]. The compressive strength of treated palm oil fuel ash (TPOFA)-based geopolymer mortar was 62.52 MPa after 120 days of curing [14].

### 1.3. Motivation for the Research

Based on the literature review, the research motivation for using u-POFA with cement mixtures is to gain insight into the potential of using u-POFA as a partial cement replacement in concrete. It is crucial to research the effects of u-POFA on the mechanical properties of concrete, the optimal dosage of u-POFA for different concrete mixtures, and the environmental benefits of using u-POFA in concrete. Similarly, there is a need to explore the

potential of using u-POFA as a pozzolanic material in other applications, such as mortars and grouts. Previous research has shown that adding u-POFA in concrete improves its mechanical properties, reduces the amount of the required cement, and reduces the carbon footprint during the concrete production process.

### 1.4. Research Objective

The first objective of this study is to explore the potential of u-POFA as a base material for alkali-activated and geopolymer products. This focused on POFA properties and the feasibility of adding POFA to alkali-activated and geopolymer products. It also investigated the effects of u-POFA on the compressive strength and microstructure of binary blends containing GBFS and FA binders. The researchers also determined the effects of binder proportions and modification of the binary-blended materials on the mechanical properties of the synthesized alkali-activated mortars and employed XRD, FT-IR, FESEM, and EDX to characterize the resulting AMMs.

## 2. Materials and Methods

### 2.1. Materials

#### 2.1.1. Palm Oil Fuel Ash

The POFA used in this research was from United Oil Palm Industry in Penang, Malaysia. The first step in preparing the POFA was drying the ash in a 105 °C oven for 24 hours to remove its inherent moisture since the POFA was placed outside the mill. The POFA was sieved using a 300-mesh sieve to remove coarser particles of the partially burnt ashes from fibers and palm kernel shells, following the work by Elbasir et al. in 2017 [15]. The POFA was then ground using a ball mill fitted with 150 steel balls and rotated at 180 rpm for eight hours to obtain 10 μm particles. Unburnt carbon was removed by heating the ground POFA in a 550°C furnace for 90 minutes. The resulting POFA was designated treated t-POFA. The fine POFA, which was designated f-POFA, was obtained by grounding the t-POFA for an additional eight hours to increase its fineness, and the f-POFA was ground for another eight hours to obtain u-POFA (ultrafine POFA) with an average particle size of 1.1 μm.

#### 2.1.2. Fly Ash

The fly ash obtained from Lafarge Malaysia Berhad (Rawang Plant, Selangor, Malaysia) followed the ASTM: C618-12a specification and was separated into two groups of low-calcium FA (Class F, CaO < 10%).

#### 2.1.3. Ground Blast-Furnace Slag

YTL Cement Technical Center in Pulau Indah, Selangor, Malaysia, provided the ground blast-furnace slag (GBFS) with an exact gravity of 2.89.

#### 2.1.4. Aggregates

The fine aggregates with a 100 percent content were from river sand and had a fineness modulus of 1.85 and a specific gravity of 2.62 under saturated and surface-dry (SSD) conditions. The ratio of sand to raw materials (FA, u-POFA, and GBFS) was maintained at 1.5.

#### 2.1.5. Alkaline Activator

The alkali activator for the mortar mixtures (BR1 to BR18) was synthesized using 10 M concentration of NaOH and sodium silicate with an initial silica modulus of 2.2 ($Ms = SiO_2/Na_2O$). It contained sodium silicate with 10 M sodium hydroxide.

#### 2.1.6. Methods for Determining Material Properties

This research first analyzed the physicochemical properties of the base materials, where a Malvern 3000 laser diffraction particle size analyzer (Grovewood Road, Malvern,

Worcestershire, WR14 1XZ, United Kingdom) was used to analyze the particle-size distributions and surface area. A Rigaku RIX3000 instrument was used to carry out X-ray fluorescence (XRF) to determine the chemical composition of the base materials, and a Zeiss SupraTM 35VP was used to investigate their morphology via FESEM.

### 2.2. Design of the Mixtures

Table 1 presents the materials for preparing the different binary blends, namely FA+GBFS, u-POFA+GBFS, and u-POFA+FA, for developing the mortars. Thus, it was necessary to mix the alkali activator solution with water and sand. GBFS, FA, and u-POFA in the raw materials had a ratio of 2.5 to the alkali activator ($Na_2SiO_3/NaOH$). All mixtures were added with a minimum of 5 wt% water of the alkali-activated mortars [16]. The literature review showed that the optimum ratio for fabricating the mortars was 1.5 for the sand/binder ratio [17]. The absolute volume method was used to obtain the raw material, sand, alkali activator, and extra water proportions for the AAMs, as presented in Table 1. To ensure consistency and reproducibility, control mortars that represented different raw materials were prepared to ensure that the mortar mixtures were developed appropriately (FA: GBFS from BR14 and BR18; u-POFA: GBFS from BR8 and BR13; and u-POFA: FA from BR1 and BR7).

**Table 1.** The mixture proportions for the binary-blended, alkali-activated mortars.

| Mix | | Solid Material (kg) | | | | Alkaline Activator | | | |
|---|---|---|---|---|---|---|---|---|---|
| | | u-POFA | FA | GBFS | Sand | $Na_2SiO_3$ (kg) | 10 M NaOH (kg) | Water (kg) | Added Water (kg) |
| | | | | | Mixture (u-POFA+FA) | | | | |
| BR1 | POFA % | 0.856 | 0.00 | 0.00 | 1.280 | 0.293 | 0.040 | 0.08 | 0.06 |
| BR 2 | 10% FA | 0.770 | 0.0856 | 0.00 | 1.280 | 0.293 | 0.040 | 0.08 | 0.06 |
| BR 3 | 20% FA | 0.685 | 0.1712 | 0.00 | 1.280 | 0.293 | 0.040 | 0.08 | 0.06 |
| BR 4 | 30% FA | 0.599 | 0.2567 | 0.00 | 1.280 | 0.293 | 0.040 | 0.08 | 0.06 |
| BR 5 | 50% FA | 0.428 | 0.428 | 0.00 | 1.280 | 0.293 | 0.040 | 0.08 | 0.06 |
| BR 6 | 75% FA | 0.214 | 0.6418 | 0.00 | 1.280 | 0.293 | 0.040 | 0.08 | 0.06 |
| BR7 | 100% FA | 0.00 | 0.856 | 0.00 | 1.280 | 0.293 | 0.040 | 0.08 | 0.06 |
| | | | | | Mixture (u-POFA+GBFS) | | | | |
| BR8 | 10% GBFS | 0.755 | 0.00 | 0.0839 | 1.260 | 0.312 | 0.04 | 0.09 | 0.06 |
| BR9 | 20% GBFS | 0.671 | 0.00 | 0.1678 | 1.260 | 0.312 | 0.04 | 0.09 | 0.06 |
| BR10 | 30% GBFS | 0.587 | 0.00 | 0.2517 | 1.260 | 0.312 | 0.04 | 0.09 | 0.06 |
| BR11 | 50% GBFS | 0.420 | 0.00 | 0.420 | 1.260 | 0.312 | 0.04 | 0.09 | 0.06 |
| BR12 | 75% GBFS | 0.210 | 0.00 | 0.6294 | 1.260 | 0.312 | 0.04 | 0.09 | 0.06 |
| BR13 | 100% GBFS | 0.00 | 0.00 | 0.8391 | 1.260 | 0.312 | 0.04 | 0.09 | 0.06 |
| | | | | | Mixture (FA+GBFS) | | | | |
| BR14 | 10% GBFS | 0.00 | 0.755 | 0.0839 | 1.260 | 0.312 | 0.04 | 0.09 | 0.06 |
| BR15 | 20% GBFS | 0.00 | 0.671 | 0.1678 | 1.260 | 0.312 | 0.04 | 0.09 | 0.06 |
| BR16 | 30% GBFS | 0.00 | 0.587 | 0.2517 | 1.260 | 0.312 | 0.04 | 0.09 | 0.06 |
| BR17 | 50% GBFS | 0.00 | 0.420 | 0.420 | 1.260 | 0.312 | 0.04 | 0.09 | 0.06 |
| BR18 | 75% GBFS | 0.00 | 0.210 | 0.6294 | 1.260 | 0.312 | 0.04 | 0.09 | 0.06 |

### 2.2.1. Alkali

This research produced 18 binary-blended trial mixtures by adding varying amounts of GBFS, u-POFA, and FA. Three of the (AAM) mixtures, namely u-POFA+FA, GBFS+FA, and u-POFA+GBFS, had seven different mixing ratios of (90:10), (70:30), (25:75), (0:100), (50:50), (80:20), and (100:0) (Table 1). The binary blends were mixed in a 4.73 L mixer for two minutes to remove air pockets, and the raw materials were blended with sand, NaOH, and $Na_2SiO_3$ for ten minutes. The mortars were cast in double layers in an oil-smeared $5 \times 5 \times 5$ cm steel mold. The samples were preserved in vinyl bags for 12 h at 25°C to

minimize moisture loss. Finally, the samples were de-molded, placed in a heat-resistant vinyl sheet wrapping, and cured in an oven at 75 °C for 24 h [10].

### 2.2.2. Analysis and Testing

The mortar samples were tested for compressive strength after 7, 14, and 28 days following the ASTM C109 (ASTM, 1999) [11]. The tests were conducted using three samples of each mortar. FESEM, FTIR, and XRD were employed to characterize the mortar samples with the optimum mix.

## 3. Results and Discussion

### *3.1. Characteristics of the Raw Materials*

#### 3.1.1. Physical and Chemical Properties of the Raw Materials

Table 2 presents the chemical composition of the base materials determined through XRF. The primary base materials were CaO, $SiO_2$, and $Al_2O_3$. All base materials, namely u-POFA (64.595%), FA (49.053%), and GBFS (36.83%), had high levels of $SiO_2$. The percentages of $Al_2O_3$ were 5.851% for u-POFA, 14.44% for GBFS, and 23.516% for FA; the percentages for CaO were 39.85% for GBFS, 9.293% for u-POFA, and 5.080% for FA. The loss of ignition (LOI) for GBFS, u-POFA, and FA was 0.60%, 2.50%, and 2.13%, respectively. Figure 1 shows the particle size of the raw materials. The median particle size for the GBFS, u-POFA, and FA was 14.20 μm, 1.1 μm, and 9.8 μm, respectively. Compared to other base materials, the u-POFA had the highest surface area of 1871 $m^2$/kg.

**Table 2.** Chemical compositions of the u-POFA, FA, and GBFS analyzed via XRF.

| Oxides (%) | SiO$_2$ | Al$_2$O$_3$ | Fe$_2$O$_3$ | CaO | MgO | P$_2$O$_5$ | K$_2$O | SO$_3$ | TiO$_2$ | Na$_2$O | LOI |
|---|---|---|---|---|---|---|---|---|---|---|---|
| u-POFA | 64.595 | 5.851 | 4.737 | 9.293 | 3.130 | 5.198 | 5.219 | 0.471 | 0.216 | 0.054 | 2.50 |
| FA | 49.053 | 23.516 | 6.422 | 5.080 | 0.698 | 1.018 | 1.309 | 0.475 | 1.121 | 0.2102 | 2.130 |
| GBFS | 36.83 | 14.44 | 0.396 | 39.35 | 3.592 | 0.0191 | 0.3761 | 4.207 | 0.402 | 0.0593 | 0.601 |

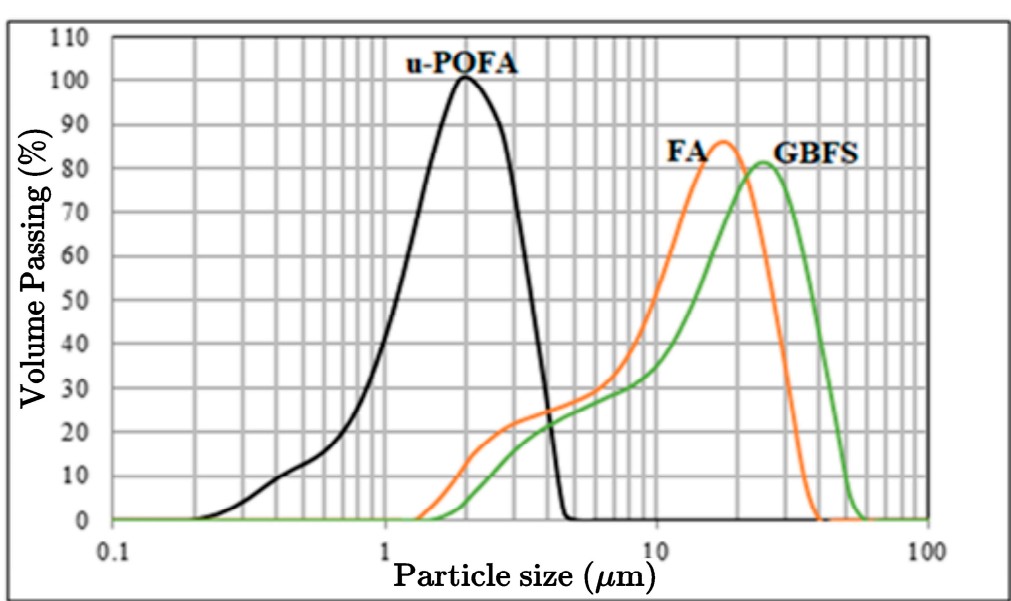

**Figure 1.** Particle-size distribution curves of the base materials.

#### 3.1.2. Mineralogy of the Raw Materials

Figure 2 shows the XRD patterns of the raw materials (GBFS, u-POFA, and FA). The GBFS displayed some amorphous stages, with broad humps between the 20–40° of 2θ° and other crystalline stages, namely spurrite $Ca_5(SiO_4)_2CO_3$, hatrurite ($Ca_3SiO_5$), quartz ($SiO_2$),

and anorthite $CaAl_2Si_2O_8$. The u-POFA primarily comprised calcite ($CaCO_3$), potassium aluminum phosphate ($K_3Al_2(PO_4)_3$), cristobalite ($SiO_2$), and quartz ($SiO_2$), while the FA was made up of sillimanite ($Al_2SiO_5$), mullite ($Al_6Si_2O_{13}$), and quartz ($SiO_2$). These results were consistent with the findings of previous studies [18,19]. Based on the chemical compositions provided by the ASTM C618 [20], POFA was a Class F mineral admixture.

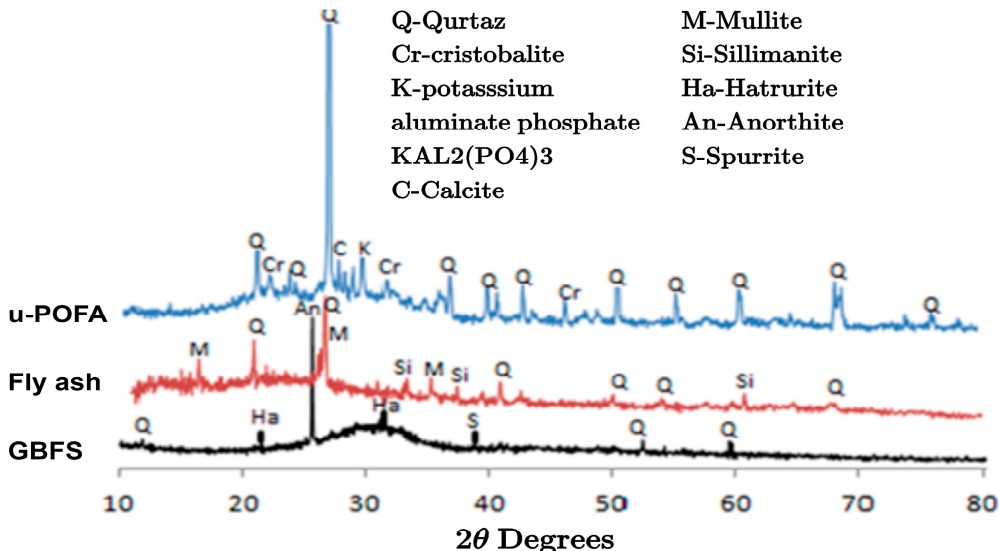

**Figure 2.** XRD patterns of the base materials (u-POFA, FA, and GBFS).

### 3.1.3. Particle Morphology of the Raw Materials

Figure 3 shows the FA, GBFS, and u-POFA particle morphology determined via FESEM. Figure 3a shows that the irregular-shaped particles of the u-POFA have irregular and porous surfaces. Figure 3b shows that the spherical particles of the FC range between 10 and 200 μm. The individual FA particle is hollow and most likely contains other smaller particles. Figure 3c shows the morphology of the GBFS, which has diamond- and square-shaped particles.

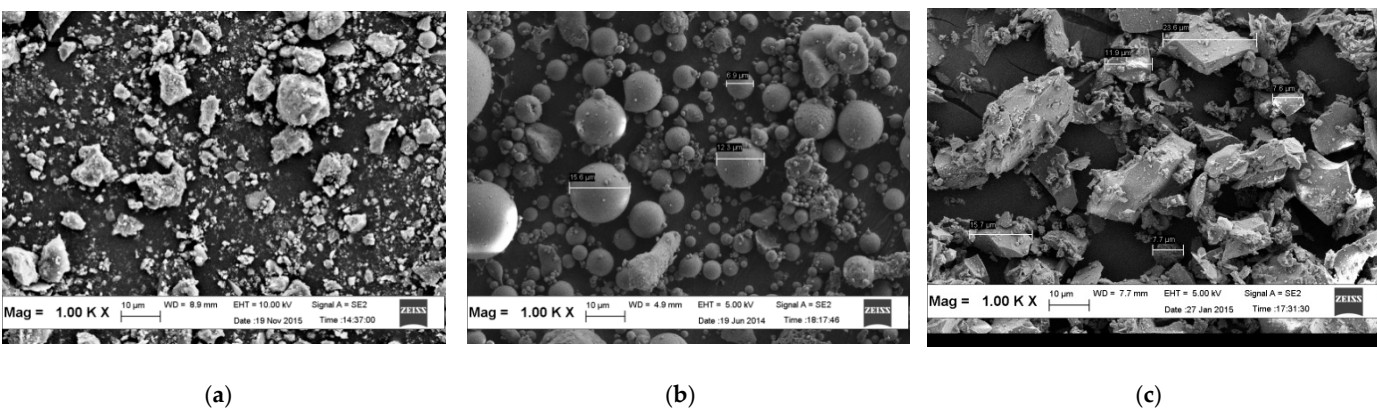

**Figure 3.** Particle morphology of (**a**) u-POFA, (**b**) FA, and (**c**) GBFS.

### 3.2. Compressive Strength

### 3.2.1. The Effects of FA on the u-POFA-Based Mortars

Figure 4 shows the effects of u-POFA, curing time (7–28 days), and the FA ratio in the alkali-activated mortars (BR1 to BR7) on the compressive strength of the fabricated mortars. The highest mortar strength for the mortar with 75 wt% FA (BR6) was 54.82 MPa at 28 days, which is comparatively more favorable than the unblended FA (BR7, 53.20 MPa at 28 days).

Adding u-POFA and FA increased the strength of the binary-blended, alkali-activated mortar. Table 3 shows that the compressive strength of the geo-polymeric system was directly influenced by the whole blended AAM with varying $SiO_2/Al_2O_3$ ratios. Compared to lower $SiO_2/Al_2O_3$ ratios, by including the BR2 to BR7, the BR1 (unblended u-POFA) had the lowest compressive strength with the most significant $SiO_2/Al_2O_3$ ratio of 11.3. As observed by other researchers [12,21], $Al(OH)_2-4$ species were permitted to condense within the mortar materials during the initial stages due to the lower Si and higher Al levels within the blended alkali-activated mortars, AAMs BR4–BR7. The phenomena allowed these mortars to build greater strength of compression than the BR1 mortar.

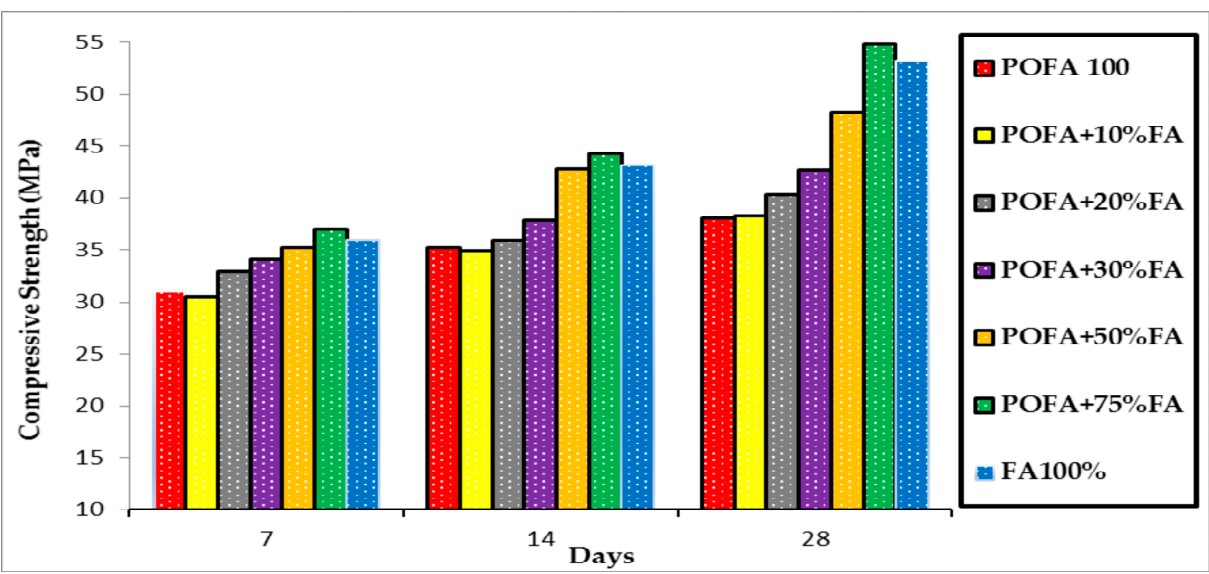

**Figure 4.** Compressive strength of the alkali-activated, binary-blended, u-POFA+FA-based mortars at 7, 14, and 28 days.

**Table 3.** The composition of some oxides in the alkali-activated mortars.

| Mix No. | | Mix Ratio (%) | $SiO_2/Al_2O_3$ | $SiO_2/CaO$ |
|---|---|---|---|---|
| BR1 | 100% POFA | 100:00 | 11.03 | 6.95 |
| BR6 | 25% POFA, 75% FA | 25:75 | 2.77 | 8.62 |
| BR12 | 25% POFA, 75% GBFS | 25:75 | 3.56 | 1.376 |
| BR7 | 100% FA | 100:00 | 2.08 | 9.65 |
| BR18 | 25% FA, 75% GBFS | 25:75 | 2.5 | 1.35 |

### 3.2.2. The Effects of GBFS on u-POFA-Based Mortars

Figure 5 shows the compressive strength of the distinct blends of u-POFA+GBFS mortars (BR8 to BR13) treated for 7 to 28 days. The maximum compressive strength of the unblended u-POFA (BR1) was 37.52 MPa after 28 days of treatment. Increasing the GBFS ratio from 10 to 75% increased the compressive strength of the binary-blended u-POFA+GBFS mortar from 21.42 to 185.6%. The BR12 with 75 wt% GBFS, cured for 28 days, had a higher compressive strength of 88.78 MPa than the unblended GBFS mortar, which had a compressive strength of 86.79 MPa. u-POFA significantly increased the strength of the alkali-activated GBFS, or geopolymer, during the blending process. GBFS is vital in enhancing the compressive strength of alkali-activated u-POFA. The higher percentages of GBFS in the mixture increased the compressive strength of the chemical-activated mortar. The change in compressive strength was consistent with the findings of other investigations [6,22]. The BR12 mixture had the highest strength (Figure 5). The unblended GBFS's compressive strength decreased without u-POFA. Compared to u-POFA, GBFS contains a significant amount of Ca and Al. As a result, the continuous rise in the mortar's achievable compressive strength is supported by Ca and Al [23]. Since GBFS serves as a viable source of Ca, it is directly associated with changes in the compressive strength of alkali-activated mortars.

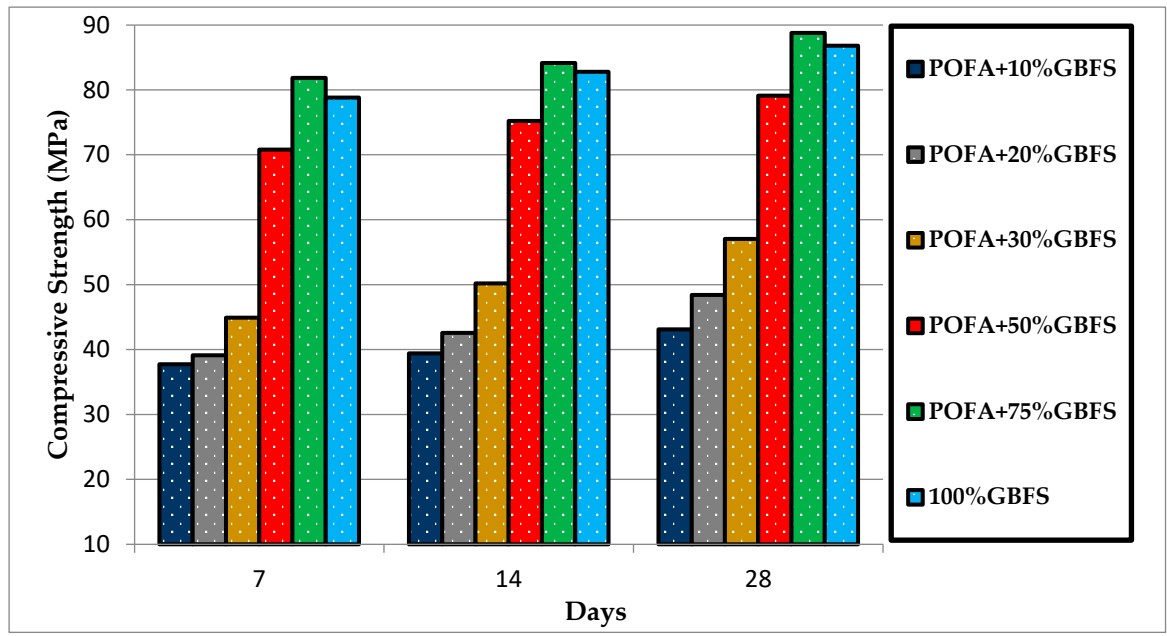

**Figure 5.** Compressive strength of the alkali-activated, binary-blended u-POFA+GBFS-based mortars at 7, 14, and 28 days.

### 3.2.3. The Effects of GBFS on FA-Based Mortars

Figure 6 shows the effects of the GBFS mortars (BR14 to BR18) on the compressive strength of the FA-based mortar. Each mortar's compressive strength raised the strength of alkali-activated, binary-blended, FA+GBFS-based mortars that were treated for a duration of 7, 14 and 28 days. A compressive strength alteration was set up by the investigation for a distinct blend of FA, with rising levels of GBFS. Compared to those treated for 7–14 days, the mortars treated for 28 days had the highest compressive strength. The maximum compressive strength of 53.88 MPa of the BR7 (unblended FA, 28 days) increased to 87.8 MPa when mixed with 75 wt% of GBFS. The binary blending of GBFS and FA significantly increased their strength. The higher compressive strength of the FA-based geopolymer was due to the higher percentage of GBFS, which was consistent with the results reported by Xu et al. [7]. The high compressive strength of the geopolymer concrete was due to the GBFS replacement ratio from 50 to 80%, subjected to wet or air curing [24].

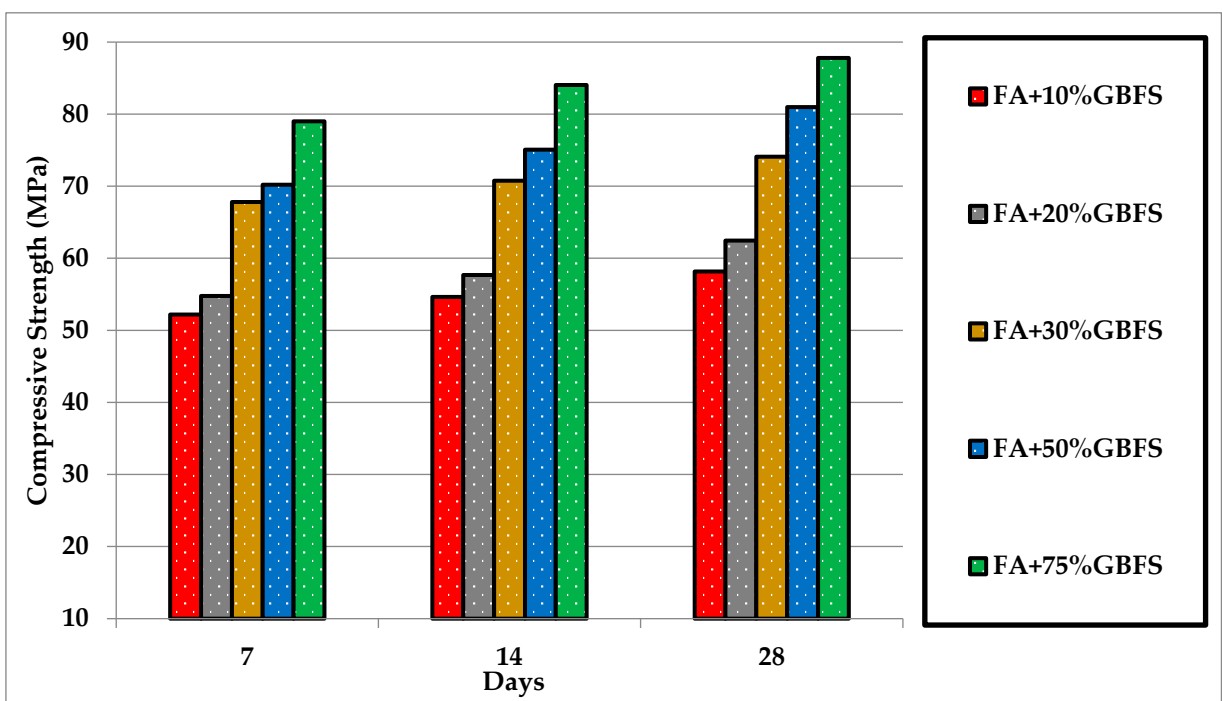

**Figure 6.** Compressive strength of the alkali-activated, binary-blended FA+GBFS-based mortars after 7, 14, and 28 days.

## 4. Characteristics of the Binary-Blended, Alkali-Activated Mortar Mixtures

### 4.1. Mineralogical Analysis

The best binary-blended mortars were (BR18; FA+GBFS), (BR6; u-POFA+FA), and (BR12; u-POF+GBFS). Figure 7 shows the XRD diffractograms for the alkali-activated mortars, namely (BR13; GBFS), (BR1; u-POFA), and (BR7; FA). The crystalline features with high-intensity peaks in the diffractograms are apparent for calcite ($CaCO_3$), albite $Na(AlSi_3O_8)$, hillebrandite ($Ca_2SiO_3(OH)_2$, mullite ($3Al_2O_3SiO_2$), analcime ($NaAlSi_2O_6 \cdot H_2O$), and quartz ($SiO_2$). The crystal stage indicates that different active reactions occurred during mortar fabrication. The primary mineral structures in the BR6 are analcime, mullite, and quarts, where analcime and mullite are the precipitations of sodium–alumina–silicate–hydrate (N–A–S–H) present in the base materials. The main stages of the BR12 are quartz, hillebrandite, and calcite. The main stages of the BR18 are albite, anorthite, and quartz. The peak of the anorthite stage and that of calcium–aluminosilicate–hydrate (C–(A)–S–H) [25] are similar. The primary stage in the BR18, BR6, and BR12 is quartz. Each mortar appeared to go through distinct reaction stages in its development, as indicated by the peak intensity of the corresponding mineral stages of the mortars [12]. The geopolymer binders are of high quality because of the presence of hillebrandite and albite in the main steps of the mortars [26,27]. The albite stages are from aluminum silicate. Si is in a tetrahedral form in one chain, and Na and Al are in the octahedral structure in their crystalline structure stages. The hillebrandite phase, $Ca_2SiO_3(OH)_2$, is an inherent stage of the $CaO$–$SiO_2$-$H_2O$ ternary system, with a high level of calcium silicate hydrate (C–S–H). The calcite ($CaCO_3$) stage in the BR13 and BR6 was produced via the reaction between $CO_2$ and $CaO$ during AAM development [28].

### 4.2. Fourier-Transform Infrared (FTIR) Spectroscopy

Figure 8 shows the FTIR vibration bands for the alkali-activated mortars (BR13; GBFS), (BR1; u-POFA), (BR7; FA), and (BR13; GBFS), and the optimal binary-blended mortar mixtures (BR18; FA+GBFS), (BR6; u-POFA+FA), and (BR12; u-POF+GBFS). Following the vibrations of Si–O and Si-O-Al or Si–O–Si in-plane and blending modes, the existence of (C–(A)–S–H) and C–S–H and N–A–S–H gels in the mortars are displayed by the

spectra for distinct vibration bands [29,30]. The symmetric stretching vibration (O–Si–O or Si–O–Si and) is within 730.95–777 cm$^{-1}$, and the AlO$_2$ functional group is within 650.12 cm$^{-1}$–695 cm$^{-1}$ [12,29]. The bands for sodium bicarbonate deformation were responsible for the peaks in the vibration bands at 874.70–876.70 cm$^{-1}$ [30,31]. The asymmetric Si–O–T (T = Al or Si) vibration within the geopolymer gels or zeolite is suggested to be the cause of the peaks of the vibration bands at 1006.72–1044.42 cm$^{-1}$ [32–34]. Figure 8 shows the spectrum for the atmospheric decarbonization of a geopolymer between bandwidth 1400 and 1500 cm$^{-1}$, which indicates the presence of O–C–O stretching vibrations. The H–O–H bending and H–O–H stretching are at 1645 cm$^{-1}$ and 3468 cm$^{-1}$ for the O–H, respectively [25]. The peaks share a likeness with the characteristic stretching vibrations of the –OH of water's hydrogen bonding integrated within the structure of a geopolymer mortar [35]. The FTIR spectroscopy showed the reaction products formed from the generation of mortars with functional groups. The time for hydration to obtain the hydrated matrixes of the mortars, which are made up of O, Ca, and Si, may be increased by the functional groups' presence.

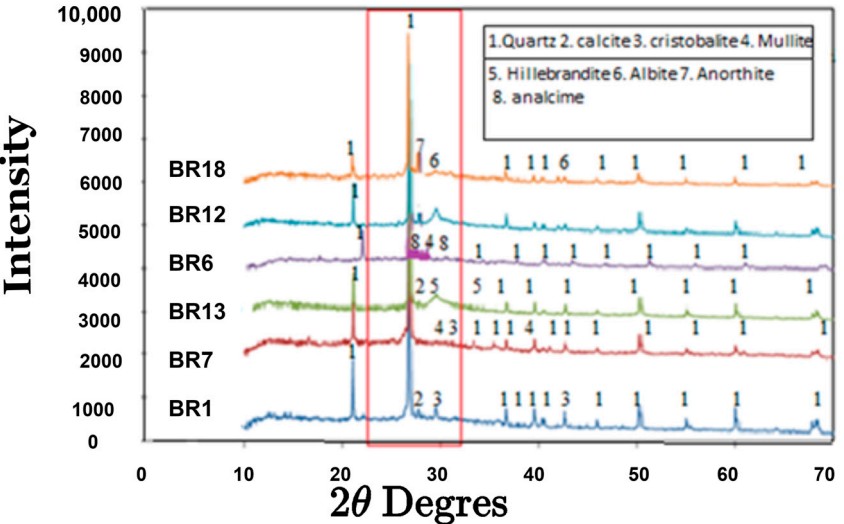

**Figure 7.** XRD of the alkali-activated mortars at 28 days.

*4.3. Field Emission Scanning Electron Microscopy (FESEM)*

Figure 9 shows the microstructures of the binary-blended, alkali-activated mortars (BR1; u-POFA), (BR7; FA), and (BR13; GBFS) and the optimal combination for the binary-blended mortars (BR6; u-POFA+FA), (Br12; u-POF+GBFS), and (Br18; FA+GBFS) after 28 days of treatment. The figure also shows the FESEM images of (BR1; u-POFA), (BR7; FA), and (BR13; GBFS) used to examine the morphological properties of the FA-, u-POFA-, and GBFS-based alkali-activated mortar samples. The FESEM micrograph for the u-POFA alkali-activated binder illustrates an enhanced look with a more compact, dense, and discretely uniformed cross section because of the higher ratio of surface area to volume, showing a comparatively smoother structure than fly ash. The microstructure of the FA ash particles shows that it contains reacted and unreacted fly ash portions, which have numerous minute pores. The micrographs feature GBFS-bonding features in the interfacial zone between the slag paste and the alkali-activated binder. A transition zone formation, which is uniform and dense, appears. There is a gradual increase in pore refinement and advancement in the matrix, and the matrix–aggregate interface can be observed to have higher GBFS contents. Hence, these results are consistent with the improvement in compressive strength that is needed to obtain the best combinations of binary-blended mortars BR6 (u-POFA+FA), BR18 (FA+GBFS), and BR12 (u-POF+GBFS). The microstructure of the BR6 primarily comprises FA, which is the congested nanofiber bulk on the surface of the unreacted FA. Silva et al. [20] showed that hydrolysis reactions occurred on the surfaces of solid particles. The BR6 has

many particles with a highly dispersed matrix of small pores.Figure 9 shows that the pores contain unreacted particles [33]. u-POFA altered the microstructure of the POFA/FA-based alkali-activated mortar, as indicated by "b" in Figure 9. The BR12 displays substantial morphological changes caused by the alkali-activated reaction between POFA (25%) and GBFS (75%). The morphology shows small cracks, and the intact particles are smooth and spherical. The BR18 contains 75% GBFS and 25% FA, and the morphology produces products with extensive alkali-activated reactions. The particles in the BR18 were from the FA, and these particles are primarily spherical and irregular-shaped GBFS particles.

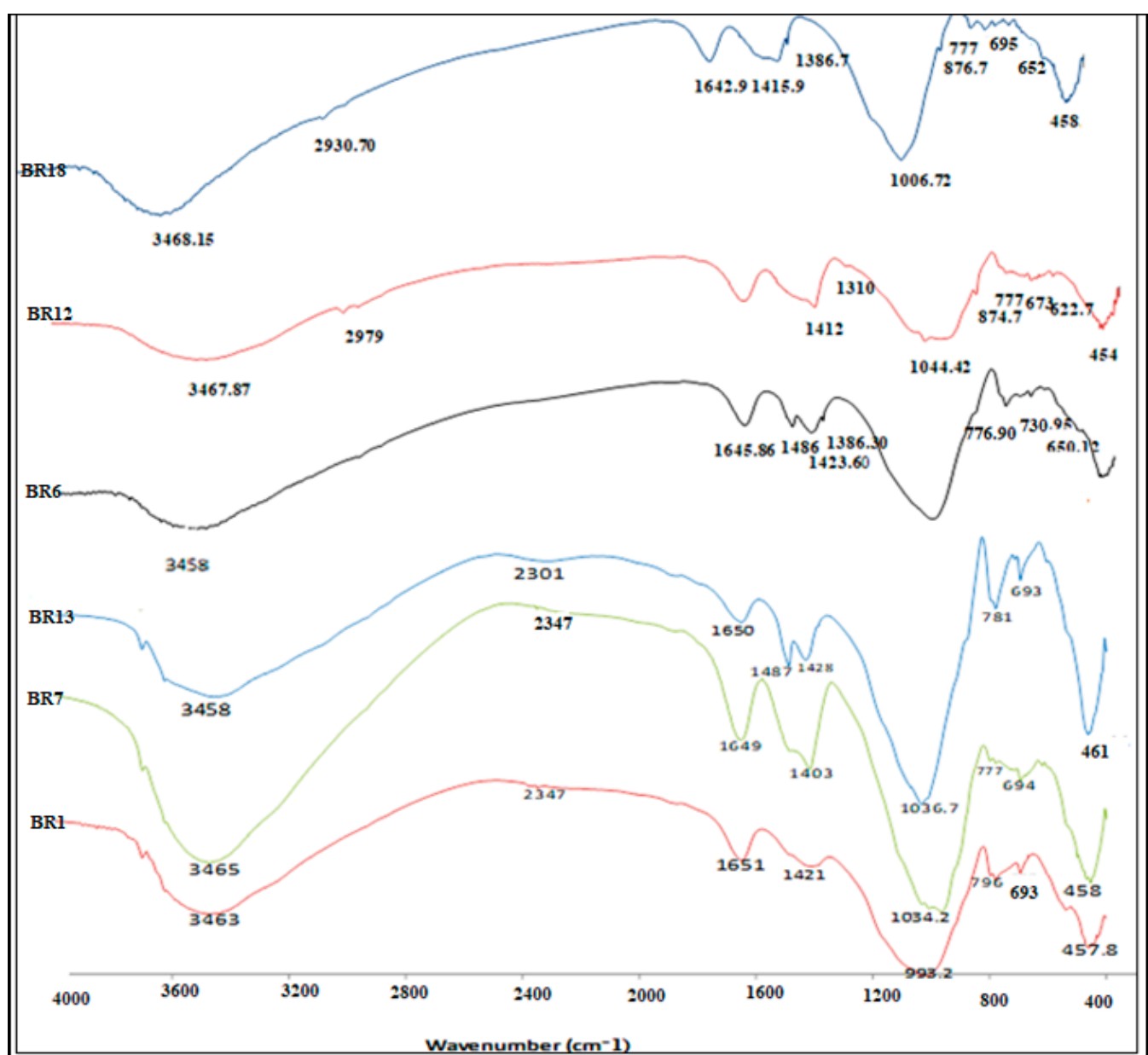

**Figure 8.** FTIR for alkali-activated mortars at 28 days.

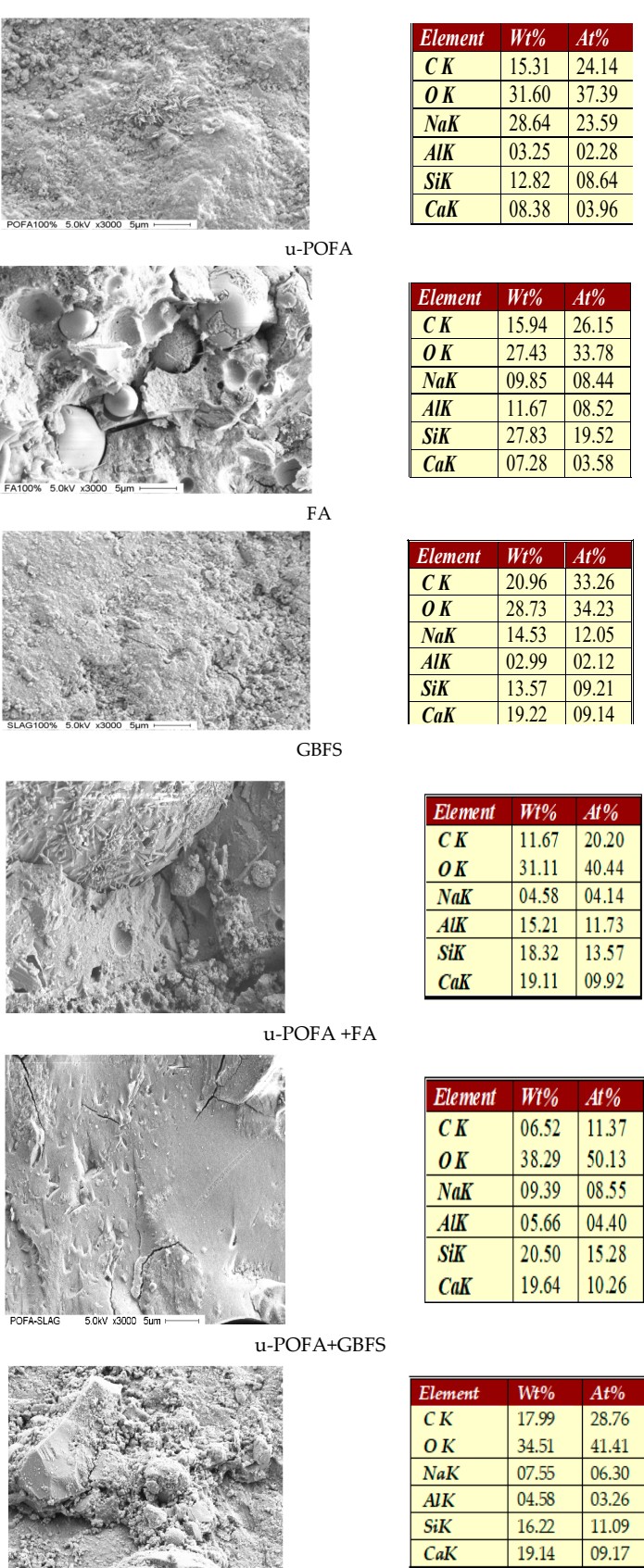

**Figure 9.** FESM for the alkali-activated mortars at 28 days.

## 5. Conclusions

The ultrafine palm oil fuel ash enhanced the strength of granulated-blast-furnace-slag- and fly-ash-based, binary-blended, alkali-activated mortars. Adding 25 wt% u-POFA enhanced the compressive strength of the GBFS and FA mortars. This enhancement was due to the optimal $CaO/SiO_2$ and $SiO_2/Al_2O_3$ ratios in the mortars, which were directly associated with the formation of calcium–silicate–hydrate (C–S–H), calcium–alumina–silicate–hydrate (C–(A)–S–H), and sodium–alumina–silicate–hydrate (N–A–S–H). It was also directly associated with the transformation of the mortars' microstructure. As such, a better-quality binary-blended AAM using the appropriate ratio of GBFS or FA may be produced through abundantly accessible POFA.

**Author Contributions:** Conceptualization, O.M.M.E. and M.A.M.J.; methodology, O.M.M.E.; software, O.M.M.E.; validation, O.M.M.E., A.M. and Z.A.A.; formal analysis, O.M.M.E.; investigation, O.M.M.E.; resources, O.M.M.E.; data curation, Z.A.A.; writing—original draft preparation, A.M.; writing—review and editing, A.M.; visualization, N.S.M.; supervision, M.A.M.J.; project administration, Z.A.A.; funding acquisition, N.S.M. All authors have read and agreed to the published version of the manuscript.

**Funding:** The authors gratefully acknowledge the Universiti Sains Malaysia for providing the financial support through the Research University (1001/PAWAM/814191) Grant Scheme for undertaking this research work.

**Acknowledgments:** The authors would like to thank the Palm Oil Industries for providing the palm oil fuel ash; YTL Cement Technical Center, Pulau Indah, for providing GBFS; and Lafarge Malaysia Berhad (Associated Pan Malaysia Cement Sdn. Bhd.) for providing FA.

**Conflicts of Interest:** The authors declare no conflict of interest.

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
