# Peer review of "The Compressive Strength and Microstructure of Alkali-Activated Mortars Utilizing By-Product-Based Binary-Blended Precursors"

_2673-3161, doi:10.3390/applmech4030046_

Round 1

Reviewer 1 Report

The purpose and literature summary of the study are well expressed, and the article presentation is nicely prepared. Additionally, the study is innovative and includes current topics. The following corrections need to be made for the publication of the study:

·       Line 25: After providing the full names of the chemical compounds "Na2SiO3" and "NaOH," they should be abbreviated.

·       Line 32: The abbreviation "(C-(A)-S-H)" should be expanded to "calcium-alumina-silicate-hydrate." • Line 32 and 359: The abbreviation "(N-A-S-H)" should be expanded to "sodium-alumina-silicates-hydrate."

·       Line 37: The hyphens between "Compressive-stength" and "Ground-blast-furnace-slag" should be removed.

·       Line 42: "Palm oil fuel ash" should be abbreviated as "POFA."

·       Line 49: The abbreviated form of "Fly Ash" should be written as "(FA)."

·       Line 57: The full form of "AAM" should also be provided.

·       Line 67 and 84: Pay attention to the notation of indices in chemical representations.

·       Line 73: What is PFA?

·       Line 84: "ultrafine POFA" should be written as "u-POFA."

·       The typographical error in Line 197 should be corrected.

·       The abbreviation "GGBFS" in Figures 1 and 2 is written incorrectly. It should be corrected to "GBFS."

·       Line 285: "alkaline activated mortar" should be abbreviated as "AAM."

·       It is stated that compressive strength measurements were repeated for 3 samples. Therefore, standard error bars should be provided in the column charts given in Figures 4, 5, and 6.

·       The abbreviation "GBS" in Table 3 is written incorrectly; it should be corrected to "GBFS."

·        Line 299: The abbreviation "BBFS" is written incorrectly; it should be corrected to "GBFS."

·       Line 356: "CS" is not specified as an abbreviation for what.

·       The image quality of Figures 7 and 8 is very low and should be enhanced.

·       Line 389: "Figure 9" is written incorrectly; it should be "Figure "

·       Figure 8: The analyses in this figure are not indicated for which mixtures (BR1, BR7, etc.).

·       Line 406: Where is "Figure 4.15"?

·       The conclusion section is too short and can be further elaborated. For example, the percentage increase or decrease in strength can be specified. Additionally, recommendations should be made for future studies.

·       In the conclusion section, "u-POFA, GBFS, and FA" should have their full forms mentioned at their first occurrence.

·       Line 434 and 442: The abbreviation "GBBFS" is incorrect; it should be corrected to "GBFS."

Author Response

Authors are thankful to all the reviewers for giving the relevant valuable comments and suggestion to improve the manuscript.

Further, authors have made appropriate changes to improve the manuscript.

Reviewer 1 comments and author responses are as below:

Line 25: After providing the full names of the chemical compounds "Na2SiO3" and "NaOH," they should be abbreviated.

Authors response:

Sodium Silicate (Na2SiO3

Sodium Hydroxide (NaOH)

Line 32: The abbreviation "(C-(A)-S-H)" should be expanded to "calcium-alumina-silicate-hydrate." • Line 32 and 359: The abbreviation "(N-A-S-H)" should be expanded to "sodium-alumina-silicates-hydrate"

Authors response

Amended as the reviewer requsted.

 Line 37: The hyphens between "Compressive-stength" and "Ground-blast-furnace-slag" should be removed.

Authors response

Amended as the reviewer requsted.

 Line 42: "Palm oil fuel ash" should be abbreviated as "POFA."

Authors response

Amended as the reviewer requsted.

Line 49: The abbreviated form of "Fly Ash" should be written as "(FA)."

Authors response

Amended as the reviewer requsted.

Line 57: The full form of "AAM" should also be provided.

Authors response

AAM = alkaline activated mortars. Authors did define it in the abstract. Done.

 Line 67 and 84: Pay attention to the notation of indices in chemical representations.

Authors response

Amended as the reviewer requsted.

Line 73: What is PFA?

Authors response: Pulverized fuel ash

Line 84: "ultrafine POFA" should be written as "u-POFA."

Authors response

Amended as the reviewer requsted.

The typographical error in Line 197 should be corrected.

Authors response

Amended as the reviewer requsted.

The abbreviation "GGBFS" in Figures 1 and 2 is written incorrectly. It should be corrected to "GBFS."

Amended as the reviewer requsted.

Line 285: "alkaline activated mortar" should be abbreviated as "AAM."

Authors response

Amended as the reviewer requsted.

It is stated that compressive strength measurements were repeated for 3 samples.

Therefore, standard error bars should be provided in the column charts given in Figures 4, 5, and 6.

Authors response

In this case we took the average results from three samples based on the specification used in this work.

The abbreviation "GBS" in Table 3 is written incorrectly; it should be corrected to "GBFS."

Authors response

Amended as the reviewer requsted.

  • Line 299: The abbreviation "BBFS" is written incorrectly; it should be corrected to "GBFS."

Authors response

Amended as the reviewer requsted.

Line 356: "CS" is not specified as an abbreviation for what.

Compressive strength  "CS"

The image quality of Figures 7 and 8 is very low and should be enhanced.

As suggested, Figures 7 and 8 are now clear enough with goog reslution .

Line 389: "Figure 9" is written incorrectly; it should be "Figure "

Amended as the reviewer requsted.

Figure 8: The analyses in this figure are not indicated for which mixtures (BR1, BR7, etc.).

Amended as the reviewer requsted.

Line 406: Where is "Figure 4.15"?

Authors response

There is no figure 4.15, it was error. Author means “Figure 9” to show the FESM results. Now it is Figure 9 in the text.

The conclusion section is too short and can be further elaborated. For example, the percentage increase or decrease in strength can be specified. Additionally, recommendations should be made for future studies.

In the conclusion section, "u-POFA, GBFS, and FA" should have their full forms mentioned at their first occurrence.

Authors response

As suggested, u-POFA, GBFS, and FA  were adjusted substantially as a reviewer's suggestion.

Line 434 and 442: The abbreviation "GBBFS" is incorrect; it should be corrected to "GBFS."

Authors response

As suggested, abbreviation "GBBFS"  is adjusted substantially as a reviewer's suggestion.

Reviewer 2 Report

Manuscript ID: applmech-2464263

Type: Article

Title: Compressive Strength Performance and Microstructure of Alkali Activated Mortars Utilizing by-Product-Based, Binary Blended Precursors

Congratulations on an interesting research experiment. This is a topic that fits in with the ideas of sustainable development. The use of waste in the technology of cement composites is still an important topic undertaken by many researchers. Transparency of the research methodology and repeatability of the obtained results are also important. The use of powder waste, which can replace cement and lead to a reduction in the carbon footprint, is an important issue. Therefore, I generally evaluate the experiment positively.

Comments:

Please specify u-POFA abbreviation when the sentence begins with this.

The density of all the ingredients used in the analyzed mortar variants is missing. The authors wrote that mortar ingredients were selected by volume, so this information is important.

line 196-197 - should be point 2.2.1 and title

Figure 1 - remove (-)

Figure 7 - bad quality, please correct

Author Response

Authors are thankful to all the reviewers for giving the relevant valuable comments and suggestion to improve the manuscript.

Further, authors have made appropriate changes to improve the manuscript.

Reviewer 2 comments and author responses are as below:

Please specify u-POFA abbreviation when the sentence begins with this.

Authors response

Amended as the reviewer requsted., for example line , 18, 42,71.

The density of all the ingredients used in the analyzed mortar variants is missing. The authors wrote that mortar ingredients were selected by volume, so this information is important.

We used the absolute volume method which is a method commonly used in the mix design then we used the raw materials by Weight

line 196-197 - should be point 2.2.1 and title

Amended as the reviewer requsted.

Figure 1 - remove (-)

Authors response. Amended as the reviewer requsted.

Figure 7 - bad quality, please correct

The figure 7 now with good resolution,